# A Stretching Force Control-Based Cyclic Loading Method for the Evaluation of Mechanical Properties of Gelation Methacrylate (GelMA) Microfibers

**DOI:** 10.3390/mi13101703

**Published:** 2022-10-10

**Authors:** Qian Liang, Xiao Yu, Xie Chen, Qiang Huang, Tao Sun

**Affiliations:** 1Intelligent Robotics Institute, School of Mechatronical Engineering, Beijing Institute of Technology, Beijing 100081, China; 2Beijing Institute of Technology, Jinggong College, Beijing 100081, China

**Keywords:** elastic modulus, GelMA microfiber, cyclic loading, stress-strain loop, force control

## Abstract

Microfluidic spun gelation mechacrylate (GelMA) microfiber has been widely utilized as a promising bioink for 3D bioprinting. However, its weak and easily tuned mechanical properties are still difficult to precisely evaluate, due to the lack of an effective stretching method. In this paper, we propose a force-control-based cyclic loading method for rapidly evaluating the elastic modulus: the *E* of the microfibers with different GelMA concentrations. A two-tube manipulation system is used to stretch microfiber with a non-destructive process. Based on the model reference adaptive control strategy, the stress response can be fitted into a sinusoidal wave when a small sinusoidal strain is automatically applied onto the microfiber. Afterwards, the maximum tensile stress and tensile stain is obtained to determine the *E*. Moreover, different stress amplitudes and frequencies are applied to form different stress-strain loops with almost same *E*. Compared with a frequently-used constant force loading method, the proposed method shows an obvious advantage in measurement accuracy, especially for low-concentration GelMA microfiber. Furthermore, the reasonableness of the measured *E* for different GelMA concentrations is confirmed by 3D cell culture experiments, and the results show the proposed method has great application potential to investigate the interaction between cell and fibrous bioink substrate.

## 1. Introduction

Gelatin methacryloyl (GelMA) is a gelatin derivative that can form covalently crosslinked hydrogels in the presence of a photoinitiator [1]. Low-concentration GelMA hydrogel provides a cell-favorable soft environment for the encapsulated fibroblasts, human mesenchymal stem cells, and endothelial cells, et al. [2,3]. Moreover, microfluidic spinning method enables GelMA hydrogel to be processed into GelMA hydrogel microfibers with high-throughput capability [4]. After integrating a microfluidic platform into 3D printer technology, a novel bioprinting method can be established for higher-order cell assembly to mimic in vivo tissues from vasculature to cartilage and bone [5,6,7]. For maintaining the stability of the printed structure, the mechanical properties (MP) of bioink should be modified according to accurate value measurements in advance [8,9]. Such measurements are usually implemented based on the GelMA hydrogel bulk block rather than the microfiber [10]. It is worth mentioning that the MP of GelMA hydrogel is easily tuned by the different processing conditions, such as UV exposure time, et al. Even with the same GelMA concentration and initiator concentration in the pre-gel solution, the exposure degree and time are significantly different for the laminar solution flow in the microfluidic channel and the solution in the macro-volume channel [11,12,13]. Moreover, other biopolymers can also be mixed with GelMA hydrogel to enhance the MP or promote cell growth [14,15]. Because the MP of bulk GelMA hydrogel is difficult to exactly describe the MP of the GelMA microfiber, it is necessary to independently measure the microfiber MP for subsequent evaluation of the printed structure stability and tissue maturation.

The ratio between the loading and the consequent deformation is described by the elastic modules: *E* to characterize the MP of GelMA hydrogel [16]. For the fiber-shaped hydrogel, the common measurement method is based on a mechanical stretcher [17,18,19]. Two ends of the microfiber are respectively fixed by two mechanical clamps, and a normal stress is applied to induce microfiber deformation. Because the loading precision is limited in dozens of micro-newton ranges, the measured microfibers usually exhibit dozens to several hundred kilopascal mechanical strength to avoid immediate tensile damage. However, previous work has shown that extremely soft environments (a few kilopascals) promote the best cellular responses, such as low-concentration GelMA [20]. For measuring the soft microfiber, a tube-tube micromanipulator system was established to provide a nano-newton actuator and sensor to handle the microfibers. In this system, two measurement strategies are provided according to two force loading modes [21,22,23,24]. One mode is based on the uniform elongation rate of the microfiber. A constant deformation speed is applied up to a tension strain beyond the break point, and *E* can be calculated by using the slope of the curve between the stress and strain in the linear viscoelastic range [23]. Another mode is dependent on dynamic loading. Multi-frequency stretching force loading method based on force control was developed to measure the storage and loss module: *E*’&*E*″ of multi-type microfibers [24]. However, these two modes still have some limitations in no-damage measurements for cell-laden microfibers. In the first mode, the stretching force should be continuously increased to form the curve between the stress and strain. The risk of breakage can be obviously increased, especially for microfiber with weak MP. In the second mode, because a phase shift between the applied stress and resultant strain should be measured additionally for characterizing viscoelasticity, which usually causes more than 10 min during each experiment to induce potential danger including cell pollution, dehydration, and even death [24]. Therefore, there still lacks a low-damage method to rapidly measure the *E* of the soft microfibers.

In this paper, we propose a cyclic loading strategy based on a two-tube manipulation system, and the small amplitude oscillatory stretch stress are automatically applied for less-damage and rapid measurement of the *E* of microfibers with different GelMA concentrations. The whole measuring process is shown in Figure 1. First, the corresponding relationship between the deflection of the sensing tube and stretching force: *F_f_* is calibrated, and the effect of microscope vision on the measure result of the *F_f_* is analyzed. Second, a scalable input adaptive control strategy is applied to allow the *F_f_* to be consistent in each cycle loading. The strain and the resulting stress signals are fitted with sinusoidal functions with phase shifts, and then elliptical Lissajous crossplots are plotted to describe the stress-strain hysteresis loop under different *F_f_* and frequencies. By analyzing these crossplots, we find that the precise evaluation for the *E* can be successfully achieved, and only depends on the minimum initial *F_f_* rather than subsequently increasing *F_f_*, and the whole measuring process can be finished in 5 cycles (<30 s). Moreover, we found that our proposed strategy facilitates the distinguished measurement for the *E* of GelMA microfibers with different concentrations, especially for low-concentration GelMA (5.5% *w*/*v*), which is difficult to achieve by traditional measuring methods based on stress-strain curve. Finally, we use further cell spreading experimentation and pore structure observation under SEM to certify the reasonableness of the measured results.

## 2. Materials and Methods

### 2.1. Microfluidic Spinning of the GelMA Microfibers

A spinning microfluidic device is composed of a coaxial needle apparatus, a transparent glass tube (length: ~5 cm, inner diameter: ~1.7 mm), and a pipette tip. The coaxial needle is inserted into one end of the glass tube, and another end of the glass tube is inserted into the pipette tip. The inserted area is sealed by an elastic adhesive tape. An amount of 1.8% *w*/*v* alginate solution and cell-laden GelMA solution are injected into the microfluidic device with the flow rate of 18 mL/h and 500 μL/h, respectively, and then a core GelMA flow surrounded by an outer alginate flow is formed in the glass tube. When the glass tube is irradiated by UV light, the GelMA flow is gelated into GelMA hydrogel microfiber, in which cells can be safely encapsulated for subsequent long-time culture [12]. The spun GelMA microfibers are ejected from the glass tube by following the ejected alginate sheath flow, and then are collected in PBS solution.

### 2.2. Two-Tube Micromanipulation System Setup for Stretching the Microfiber

Uniaxial stretching of the microfiber is implemented by a two-tube micromanipulation system, as shown in Figure 1. The system consists of two subsystems: an execution system and a vision system. In the execution system, a moving and a sensing tube are installed in parallel to a motorized manipulator (3 DOF) and a manual manipulator (3 DOF), respectively. Two ends of the microfiber are hung and fixed onto two tubes, and the microfiber is immersed in PBS solution. The majority of automatic manipulator components comprise three actuators (NSA12, Newport, Irvine, USA., minimum incremental motion is 0.2 μm). They are controlled by their matched controllers (NSA-PP, Newport, Irvine, USA ), which receive the command from a PC directly. The manual manipulator (LDV50-LM-C2, SELN, Dongguan, China ) is used to adjust the position of the tube and microfiber roughly. The vision system provides feedback information about the force applied on the microfiber. It comprises a stereoscopic microscopy (SZX-16, Olympus, Tokyo, Japan ) with a digital camera (DP21, Olympus, Tokyo, Japan, maximum resolution: 1600 × 1200) that is linked to a PC to capture real-time images. There is an outside light source to improve the tube’s identification by adjusting the light intensity.

### 2.3. Calculation of the Stretching Force Applied on the GelMA Microfiber

The fixed microfiber is axially stretched by moving the manipulating tube, and then the sensing tube can show the applied stretching force: *F_f_* due to its easy-to-occur characteristic of small deflection deformation. The Euler–Bernoulli cantilever model is used to depict the relationship between the tube tip’s deflection *u_tip_* and *F_f_*. We can obtain:(1)$$u_{tip}=-\frac{F_{f}x_{a}{}^{2}}{6E_{t}I_{z}}(3L-x_{a})$$
where *u_tip_* is the deflection deformation of the tube’s tip, *E_t_* is elastic modulus of the sensing tube, *I_z_* is the moment of inertia, *L* is length of the tube, *x_a_* is taken as the coordinate along tube long axis of the left end as 0 and the right end as *L*. Because microfiber was fixed by the glue, *L* and *x_a_* is constant and can be measured before manipulation. We define a constant *K* to represent the “stiffness” of the tube. The *K* is confirmed from Equation (1) after microfiber is glued on the fixed position:(2)$$K=\frac{F_{f}}{u_{tip}}=\frac{6EI_{z}}{x_{a}{}^{3}-3Lx_{a}{}^{2}}$$

After that, the *F_f_* is directly calculated by detecting the deflection deformation *u_tip_* through:(3)$$F_{f}=Ku_{tip}$$

### 2.4. Measurement of the Microfiber’s Elastic Modulus Based on Cyclic Loading

A computer vision is utilized to provide feedback information for measuring *u_tip_*. The tip is recognized from the original picture by the following machine vision processes. First, a binarization processing is utilized to segment the profile of the sensing tube from the origin pictures. The Harris corner detection algorithm is used to search the sensing tube’s tip in the whole image to obtain the coordinates of the tip, (*x_t_*, *y_t_*). Because there are many noises after binarization to disturb the result of feature point recognition, the gray morphology operation, dilate, is used to minimize the disturbance. Then, the *u_tip_* is calculated by the difference between the initial position of the tube’s tip (*x_o_*, *y_o_*) and (*x_t_*, *y_t_*). Considering it is a small deflection, the deformation *u_tip_* could be regarded as $u_{tip}=\left|x_{t}-x_{o}\right|$ directly, and the *F_f_* is further calculated from Equation (3).

For measurement of the microfibers’ elastic modulus *E*, the cyclic movement of the manipulating tube is automatically implemented by control of the *F_f_ (t)* to track a time-dependent sinusoidal force signal. A model reference adaptive control (MRAC) method is proposed to improve the transient response of the controller, making the force control robust. The cross-sectional area of microfiber: *S* can be measured in advance, and the stress σ can be calculated by:(4)$$\sigma =\frac{F_{f}}{S}=\sigma_{0}\sin(\omega t+\delta )$$

Meanwhile, the initial length of microfibers: *L_o_* can be measured. The displacement of automatic manipulator: *L_m_* can be obtained from the motorized manipulator directly, so the strain of the microfiber: *ε* is calculated as:(5)$$\varepsilon =\frac{L_{m}-u_{tip}}{L_{o}}=\varepsilon_{0}\sin(wt+\varphi )$$

Taken together, the *E* is derived from Equations (4) and (5), and gives:(6)$$E=\frac{\sigma_{0}}{\varepsilon_{0}}$$

### 2.5. 3D Cell Culture in the GelMA Microfiber and cell Viability Staining

NIH/3T3 fibroblasts are harvested from a petri dish. Cell suspension is prepared by mixing fibroblasts and GelMA solution. To maintain good cell viability, GelMA should be dissolved in DMEM to form GleMA solution. For long-time 3D cell culture, cell-laden GelMA microfibers are immersed in DMEM kept at 37 °C with 5% CO_2_. For cell viability staining, cell-laden microfiber are immersed into 4 mL PBS solution containing 8 μL calcein-AM and 6 μL PI solution. After half an hour incubation, live and dead cells can be respectively identified by green and red points observed under inverted fluorescence microscope.

## 3. Results and Discussion

### 3.1. Accuracy Analysis for Visual-Based Evaluation of the Stretching Force

The stretching force: *F_f_* is applied along the long axis direction of microfiber immersed in PBS solution by moving the manipulating tube away from the sensing tube, and subsequently the sensing tube is slightly bent by the stretched microfiber (Figure 2a). The bending deflection deformation is measured by the microscope to obtain the *F_f_*. Specifically, the deflection at the tube tip: *u_tip_* is proportionally converted to the *F_f_* at some position of the sensing tube body: *x_a_* according to a spring stiffness *K*. Therefore, the *E* evaluation accuracy is affected by the theoretical calculation of *K* and the visual recognition for the *u_tip_*.

Firstly, we used a high-precision electronic scale (JJ224BC,G&G MEASUREMENT PLANT, Changshou, China, resolution: 0.1 mg) to calibrate the *K*, as shown in Figure 2b. A tri-prism is placed on the horizontal surface of the scale. The sensing tube is installed on a high-precision manipulator, and its neutral axis is held parallel to the tri-prism base at the initial. When the sensing tube is descended to contact with the tri-prism tip, a point-shaped contact area is generated to mimic the contact area between the sensing tube and microfiber. We can precisely obtain the deflection: *u_a_* based on the moving distance of the manipulator and the *F_f_* indicated by the scale. Based on Equation (2), we can obtain Equation (7) about *K_a_* at the position from of *x_a_*:(7)$$K_{a}=\frac{F_{f}}{u_{a}}-\frac{6EI}{x_{a}{}^{2}(3x_{a}-x_{a})}$$

Compared with the deflection of the *u_a_*, the *u_tip_* is more clearly and easily recognized by the microscope, but it is difficult to measure in this calibration system, as shown in Figure 2b. Therefore, we further combined Equations (1) and (3) to calculate the *K* dependent on the *K_a_*, as shown in Equation (8):(8)$$K=\frac{2x_{a}}{3L-x_{a}}K_{a}=\frac{2x_{a}}{3L-x_{a}}\cdot \frac{F_{f}}{u_{a}}$$

From Equation (8), the *K* responding to different *x_a_* is shown in Figure 1b, and the fitting curve was further achieved to generate the equation:(9)$$K=-\frac{6a}{213x_{a}{}^{2}-x_{a}{}^{3}}$$
in which, a is fitting parameter, and the fitting result fits well with the experimental data where the goodness of fit is 96.6%, as shown in Figure 2b. When the adhered position is fixed, the responding *K* can be obtained to calculate the *F_r_* based on the: *u_tip_*.

Furthermore, the *u_tip_* is automatically measured by detecting the position change of the characteristic corner at the tip of the sensing tube. However, because the microscope visual feedback information is easily disturbed by the inevitable fluctuation of PBS solution, the coordinates of the recognized tip corner could fluctuate to generate a random error for the *u_tip_*. To evaluate the effect of such an error on the *u_tip_*, time-dependent pixel fluctuation of the tip corner is recorded, as shown in Figure 2d. Based on the Bessel equation, the standard deviation of pixel fluctuation was calculated, and its value is 2.619 pixel. Moreover, the employed microscope (DP21, Olympus, Tokyo, Japan ) has a resolution of 1600 × 1200 responding to the field of view of 7.04 mm × 5.28 mm, therefore, each pixel represents the length of 4.4 μm. Because the visual information is provided under Objective Lens (4×), therefore, the length indicated by each pixel should be further divided by 4 * 0.5 to obtain 2.2 μm. Therefore, besides the system error induced by the fitting error of the K, there is still a random error induced by the fluctuation of the tip corner to generate ±2.8809 μm for the *u_tip_*. Manual manipulation for hanging microfiber allows *x_a_* to be larger than 60 mm, therefore, the visual measurement error for the *F_f_* is usually less than 2μN, theoretically.

### 3.2. Design and Validation of MRAC Strategy for the Stretching Force Control

Model reference adaptive control (MRAC) is utilized to achieve force-controlled stretching for microfibers. The MRAC consists of a baseline controller and an adaptive law. The baseline controller is designed based on the mathematical microfiber model. However, the model parameters are uncertain for the microfibers with different GelMA concentrations, which can affect the performance of the baseline controller. The adaptive law is further added to compensate for these parametric uncertainties and improve the performance of the baseline controller; therefore, the *F_f_* remains stably controlled by self-adjustment through the adaptive law [25]. The schematic block diagram of the MRAC is illustrated in Figure 3a.

The reference model of MRAC should be established first for the subsequent design of the baseline controller. The reference model is derived from the standard linear solid (SLS) model [26], as shown in Figure 3b, to depict microfiber viscoelasticity. The SLS model is composed of two elements: one is a spring and a dashpot connected in serial, and the other is an isolated spring connected in parallel. The SLS model is described as:(10)$$F=\frac{S}{L_{0}}(K_{2}+K_{1}e^{-t/\left(\eta /k_{1}\right)})u_{0}=(\lambda_{1}+\lambda_{2}e^{-t/\tau })u_{0}$$
where *K*_1_, *K*_2_, η are illustrated in Figure 3a, *F_f_* is the force applied on the microfiber, *L*_0_ is the initial length of the microfiber, *S* is the area of the microfiber, and *u* is the displacement of the manipulator, which is also equivalent to the deformation of microfiber. These parameters of Equation (6) can be fitted from the stress relaxation experiment of the microfiber.

The experiment data fitted by Equation (10) are shown in Figure 3b, and are listed in Table 1. The goodness of fit was 0.948, which meant the data was fitted well by the SLS model. In fact, we did not need to know the exact value of *K*_1_, *K*_2_, η, etc., though we can perform some mathematical operations to use the linear combinations of the *τ*, *λ*_1_ and *λ*_2_ to get the state space equations directly.

To control the output *F_f_* to track the force command *F_cmd_*, we employ the integral action [27] and define an integral state $e_{yI}\left(t\right)$ as:(11)$$e_{yI}=\frac{F_{f}-F_{cmd}}{s}$$
where *s* is the Laplace integral operator. Then an augmented theoretical state space equations are defined as:(12)$$\underset{\dot{x}}{\underset{︸}{\left(\begin{array}{c} {\dot{e}}_{yI}\\ {\dot{x}}_{p} \end{array}\right)}}=\underset{A}{\underset{︸}{\left(\begin{array}{cc} 0 & 1\\ 0 & -\frac{1}{\tau } \end{array}\right)}}\underset{x}{\underset{︸}{\left(\begin{array}{c} e_{yI}\\ x_{p} \end{array}\right)}}+\underset{B}{\underset{︸}{\left(\begin{array}{c} \lambda_{1}+\lambda_{2}\\ -\frac{\lambda_{2}}{\tau } \end{array}\right)}}(u+f)+\underset{B_{ref}}{\underset{︸}{\left(\begin{array}{c} -1\\ 0 \end{array}\right)}}F_{cmd}$$
(13)$$\underset{y}{\underset{︸}{\left(\begin{array}{c} e_{yI}\\ F_{f} \end{array}\right)}}=\underset{C}{\underset{︸}{\left(\begin{array}{cc} 1 & 0\\ 0 & 1 \end{array}\right)}}\underset{x}{\underset{︸}{\left(\begin{array}{c} e_{yI}\\ x_{p} \end{array}\right)}}+\underset{D}{\underset{︸}{\left(\begin{array}{c} 0\\ \lambda_{1}+\lambda_{2} \end{array}\right)}}(u+f)$$
in which, *f* is the parametric uncertainties.

However, the state *x* cannot be measured from external sensors directly, so the Luenberger observer [28] was utilized to construct the state *x* while we ignored *f*, such that:(14)$$\left\{\begin{array}{c} {\dot{x}}_{ro}=Ax_{ro}+Bu+B_{ref}y_{cmd}+L(y-y_{ro})\\ y_{ro}=Cx_{ro}+Du \end{array}\right.$$
in which, *L* is the observer gain vector and its desired value can be calculated by using Ackermann’s formula. $x_{ro}$ is the reference state constructed from the observer.

In addition, the baseline controller can be design from Equations (12) and (13) independently without adding the Luenberger observer according to the separation principle. To simplify the controller, we used the linear quadratic regulator (LQR) method with proportional feedback connections to design the baseline controller. We separated the input, *u*, into two parts as:(15)$$u=u_{bl}+u_{ad}=-K_{x}x_{ro}+u_{ad}$$
in which, $u_{bl}=-K_{x}x_{ro}$ is the output of the baseline controller, *K_x_* is the gain of the LQR and is chosen based on the optimal control theory.

Then we substituted Equation (15) into Equation (13) and let $u_{ad}=0$, so the reference model can be obtained like below:(16)$$\left\{\begin{array}{c} {\dot{x}}_{ro}=\underset{A_{ref}}{\underset{︸}{[A-LC-(B-LD)]K_{x}^{⊤}}}x_{ro}+B_{ref}y_{cmd}+Ly\\ y_{ro}=(\underset{C_{ref}}{\underset{︸}{C-DK_{x}^{⊤}}})x_{ro} \end{array}\right.$$

The baseline controller was designed and calculated based on the reference model. However, considering the uncertainties *f* existed, we only estimate the value of $K_{x}^{⊤}$ (defined as ${\hat{K}}_{x}^{⊤}$) rather than its ideal value. The estimation error ${\tilde{K}}_{x}^{⊤}$ is defined like this:(17)$${\tilde{K}}_{x}^{⊤}={\hat{K}}_{x}^{⊤}-K_{x}^{⊤}$$

At this time, the adaptive law is designed to maintain the performance of the baseline controller by reducing the estimation error ${\tilde{K}}_{x}^{⊤}$.

To calculate the adaptive law, the actual state space model also needs to be constructed. Similar to Equation (10), we build the actual model with the *f* and the actual state $\bar{x}$ like this:(18)$$\left\{\begin{array}{c} \dot{\bar{x}}=\underset{A_{ref}}{\underset{︸}{[A-LC-(B-LD)K_{x}^{⊤}}}]\bar{x}+\left(B-LD\right)(u_{ad}-{\tilde{K}}_{x}^{T}\bar{x}+f)+B_{ref}y_{cmd}+Ly\\ \bar{y}=(\underset{C_{ref}}{\underset{︸}{C-DK_{x}^{⊤}}})\bar{x}+D(u_{ad}-{\tilde{K}}_{x}^{T}\bar{x}+f) \end{array}\right.$$

The adaptive law is commonly designed from the Lyapunov’s direct method, so a Lyapunov’s function that contains the state error $e_{x}(t)=\bar{x}-x_{ro}$ should be constructed. Before that, we calculate the first derivative of $e_{x}(t)$, gives:(19)$${\dot{e}}_{x}\left(t\right)=A_{ref}e_{x}+\left(B-LD\right)(u_{ad}-{\tilde{K}}_{x}^{⊤}\bar{x}+f)$$

However, the estimation error ${\tilde{K}}_{x}^{⊤}$ and the uncertainties *f* are unknown, we assemble the ${\tilde{K}}_{x}^{⊤}\bar{x}$ and *f* into a new linear combination: $\Theta^{⊤}\Phi \left(x_{ro}\right)=-{\tilde{K}}_{x}^{⊤}\bar{x}+f$ which can be approximated by the Chebyshev polynomials. Thus the $u_{ad}$ was chosen as $u_{ad}={\hat{\Theta }}^{⊤}\Phi \left(x_{ro}\right)$, and the estimation error of ${\tilde{\Theta }}^{⊤}$ is also defined as ${\tilde{\Theta }}^{⊤}={\hat{\Theta }}^{⊤}-\Theta^{⊤}$. At this time, we consider a quadratic Lyapunov function candidate, such as:(20)$$V\left(e_{x},\tilde{\Theta }\right)=e_{x}^{⊤}P_{ref}e_{x}+2tr\left({\tilde{\Theta }}^{T}\Gamma^{-1}\tilde{\Theta }\right)\geq 0$$
where $V(e_{x},\tilde{\Theta })$ is positive definite, $\Gamma =\Gamma^{⊤}>0$ is the rates of adaptation and $P_{ref}=P_{ref}^{⊤}>0$ is the unique symmetric positive-definite solution of the algebraic Lyapunov equation.

Time-differentiating Equation (20), gives:(21)$$\dot{V}\left(e_{x},\tilde{\Theta }\right)=-e_{x}^{⊤}Q_{ref}e_{x}+2tr\left\{{\tilde{\Theta }}^{⊤}\left[\Gamma^{-1}\dot{\tilde{\Theta }}-\Phi e_{x}^{⊤}P_{ref}\left(B-LD\right)\right]\right\}$$

Because the mechanical properties of the microfiber can be considered to stay constant or change very slowly, we can consider $\dot{\hat{\Theta }}\approx \dot{\tilde{\Theta }}$. In order to ensure the stability of system during force control, $\dot{V}(e_{x},\tilde{\Theta })$ should be negative definite, so the adaptive law was chosen as:(22)$$\dot{\hat{\Theta }}=\Gamma \Phi e_{x}^{⊤}P_{ref}\left(B-LD\right)$$

The force command tracking performance can be proved to be achievable because the system is bounded-input-bounded-output (BIBO).

Simulations are implemented to test the viability of the MRAC in MATLAB. The parameters used in the simulations were adopted from the stress relaxation experiments of the microfibers with different GelMA concentrations. The parameters used for simulation are listed in Table 2.

We used $y_{cmd}=5\sin(2\pi \cdot 0.01\cdot t)+5$ to mimic the cyclic loading used for testing the force command tracking ability of MRAC, and the random noise is also added to test the system’s robustness, whose mean value was 0.277 μN. The simulation results are illustrated in Figure 3c. The experiments show that the errors between the force command and the calculated output are all less than 1 μN with different GelMA concentrations. This meant the force command tracking can be well achieved by the MRAC, and it is applicable for the microfibers with different mechanical properties. Thus, the cyclic loading applied on the microfibers can be realized with the MRAC in the next experiment.

### 3.3. Measurement of Elastic Modulus: E_c_ of GelMA Microfibers

The microfiber used in this experiment is made from 7% *w*/*v* GelMA with the diameter of ~150 μm. The microfiber is stretched by moving the manipulating tube away from the sensing tube, and then the zero position of the manipulating tube is specially set when the visual feedback system begins to detect the slight increase tendency of the stretching force *F_f_*. Afterwards, a cyclic loading is implemented by making the *F_f_* varied, followed with a preset sinusoidal force signal: *F_cmd_*. Although the variation and frequency of the *F**_f_* signal almost keeps consistent with the preset signal, there is an obvious phase delay between them. In addition, considering that the *F_f_* signal is directly converted from the *u_tip_*, ambient noise is inevitably present to induce signal fluctuation, as shown in Figure 4a. Fluctuation-induced signal error is periodically generated for different amplitudes and frequencies, as shown in Figure 4b,c, respectively.

Because *u_tip_* (~10 μm) is much smaller than the moving distance of the manipulating tube (several hundred micrometers), microfiber strain along its long axis direction could be considered to be same with the displacement data provided by the motorized manipulator. The response of the microfiber strain to the stretching stress is depicted by the scatter plot, as shown in Figure 5a. We find the distribution of these data points has an elliptic form. Furthermore, we fit both the stress and stain signals with sinusoidal functions with phase shifts, and Lissajous curve is established from the fitted signals to describe the stress-strain loop, as shown in Figure 5a. The Lissajous curves consist of plots of the intracycle periodic stress normalized for stress amplitude, plotted against the strain date normalized for strain amplitude, and the curves are elliptical in the linear viscoelastic region. According to small-amplitude oscillatory shear theory, the elastic modulus *E_c_* is the ratio of maximum stress and maximum strain. Corresponding to the Lissajous crossplot, we use the crossover point A to substitute the point B responding to the maximum strain to calculate the: *E_c_* because of the relatively large ratio of the long and short axis of the resulting ellipse. The long axis of the ellipse can project to the *x* and *y* axis to form the length: *L_x_* and *L_y_*, and the *E_c_* can be directly calculated by the ratio of *L_y_* to *L_x_*. Furthermore, we use different *F_f_* to cyclically stretch the microfiber under the same frequency to form different elliptical stress-strain loops. The values of the measured *E_c_* are almost the same for different *F_f_*, and even further for different frequency with the same *F_f_*, as shown in Figure 5b,c. Higher frequency induces a larger area of the loop indicating higher energy dissipation.

To verify the reasonableness of such cyclic measurement results, we further used a traditional constant force loading method to measure the *E_cf_* as the control group. We set the velocity of the manipulating tube at 10 mm/min, and continuously increase the stretching force: *F_cf_* to the respected values, as shown in Figure 5d. The stretching process is sampled with a time interval of 0.2 s to establish the stress-strain scatter plot. The stress-strain changing trends are almost consistent for four different *F_cf_*, and the viscoelasticity-induced delay between the stress and strain is clearly observed at the initial and final phase of the stretching process. Differing from the ellipse-shaped phase lag, a horizontal and vertical line-shaped phase lag is formed because of the lack of unloading process. When the stretched microfiber begins to be elongated at the initial phase, the *F_cf_* has a delay in being transmitted from the microfiber to the sensing tube. Similarly, the *F_cf_* still increases when the manipulating tube has stopped to stretch the microfiber at the final phase. We use a linear equation to fit the stress and strain, and the resulting *E_cf_* is the same as the *E_s_* measured in the cyclic loading, as shown in Figure 5d. Such a similarity shows the increased *F_cf_* could be substituted by a small and constant *F_f_* to achieve the *E* measurement. Moreover, practical experience shows the measurement can be simply achieved as long as the *F_f_* is sensed by the sensing tube, though it is time-consuming to get enough data to form a linear stress-strain curve whose slope is available to be utilized to calculate the *E*.

### 3.4. Evaluation of Measurement Reasonableness Based on Cell Response in 3D

Based on the same microfluidic spinning condition, NIH/3T3 fibroblasts are respectively encapsulated into 5.5% and 8.5% *w*/*v* GelMA microfibers for 3D cell culture. At initial phase, high cell viability is shown in both microfibers because of the mild spinning environment. After 10 days of culture, cells randomly spread in 5.5% GelMA microfibers, while cell aggregations generated in 8.5% *w*/*v* GelMA microfibers, as shown in Figure 6a. Because high GelMA concentration can provide sufficient cell adhesion sites for cell spreading, we conjecture that such aggregation behavior is not induced by lack of cell adhesion sites as in alginate microfiber [29], which is also proved by a cellular layer that is stably formed on the surface of 8.5% *w*/*v* GelMA microfiber. For the same initiator concentration and spinning conditions, the average pore sizes in GelMA hydrogel are inversely related to the concentration of GelMA, and mechanical properties can be correspondingly tuned for the resulting GelMA microfiber [9]. Therefore, we consider that different cell morphologies are also related to tuned mechanical properties.

Figure 6c shows the average values of the measured *E_c_* and *E_cf_* of two microfibers under a cyclic and constant force loading method, respectively. The *E_c_* for different concentrations can be distinguished under cyclic loading. Such measurement results are well corresponding to cell responses in two GelMA microfibers. High *E_c_* for 8.5% *w*/*v* GelMA concentration indicated dense pore size distribution to form high density of cell adhesion site to prevent from cell spreading, while relatively large pores with the low site density allow cell spreading in 5.5% *w*/*v* GelMA microfiber with low *E_c_*. Cell response on the microfiber is almost consistent with the bulk hydrogel with the same GelMA concentration. According to the standard data measured from the bulk hydrogel (Suzhou Intelligent Manufacturing Research Institute, Suzhou, China), the *E* of 5% and 10% *w*/*v* GelMA hydrogels are 16.63 and 61.84 kPa, respectively, but the *E_c_* of microfibers for 5.5% and 8.5% *w*/*v* are less than 10 kPa even though material-related parameters are identical. Therefore, there is an obvious difference for the *E* between hydrogel microfiber and bulk hydrogel, and such a difference may be induced by the exposure time. Furthermore, for constant force loading, *E_cf_* for 5.5% *w*/*v* is obviously smaller than 7% *w*/*v*. We further analyze the reason for such obvious differences, as shown in Figure 6c. We found the *E_cf_*-indicated slope of the fitted curve is strongly related to sample size. The five samples may be gathered in both the linear and non-linear viscoelastic ranges. Therefore, when the curve is fitted dependent on the five samples (black line), the samples in the non-linear range obviously enable the slope to be reduced. Moreover, *E_cf_* can increase when only three previous samples are utilized to be fitted (red line), while *E_cf_* is larger than that for 8.5% *w*/*v* according to the curve fitted by the previous two samples (blue line). This above-mentioned result further shows the cyclic loading method can provide a sensitive approach to precisely measure the E of microfibers. Because a weaker stretching force is required in the cyclic method relative to the traditional method, and the force-induced deformation in a time-dependent manner could be reduced to further suppress the measurement error to facilitate precise measurements. In addition, our proposed method can also be used to investigate microfiber viscoelasticity because of dynamic loading process, and viscoelastic behaviors are critical for re-establishment of cell microenvironment [30,31].

## 4. Conclusions

Elastic modulus: *E* is a basic indicator to describe the mechanical complexity of biological tissue. We propose a force-controlled cyclic stretching method to evaluate the *E* of the GelMA microfibers with characteristics of high stability, high precision and low-damage. Therefore, our proposed method can promote the investigation of the effect of mechanical properties on cell behaviors, the study of interaction between cells and substrate, and microfiber material design for 3D bioprinting.

## Figures and Tables

**Figure 1 micromachines-13-01703-f001:**
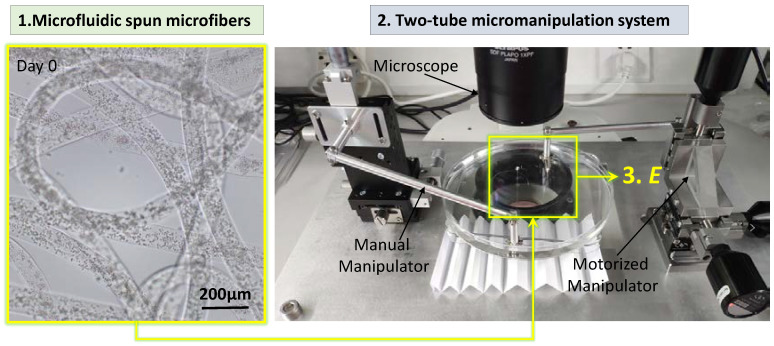
Measurement process. Cell-laden GelMA microfibers are first spun by microfluidic device, and the tow-tube micromanipulation system is utilized to measure the microfiber *E*.

**Figure 2 micromachines-13-01703-f002:**
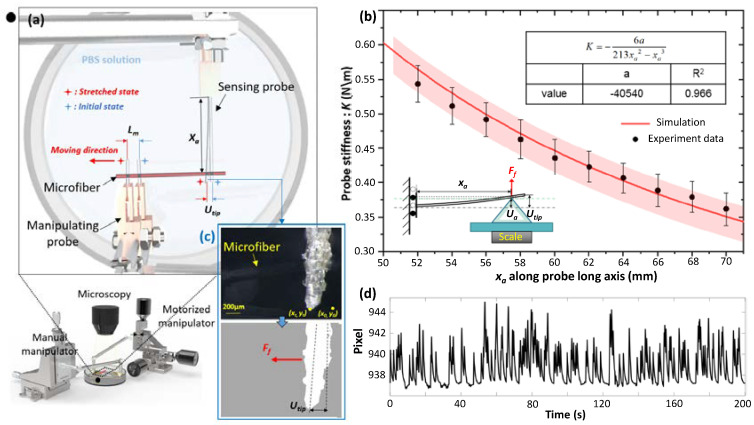
Measuring accuracy analysis of the sensing tube-indicated stretching force. (**a**) Schematic of two−tube stretching manipulation. (**b**) Calibration of the stiffness: *K* of the sensing tube. (**c**) Process of the sensing tube’s tip recognition to measure the *u_tip_*. (**d**) Pixel fluctuation process of the tip.

**Figure 3 micromachines-13-01703-f003:**
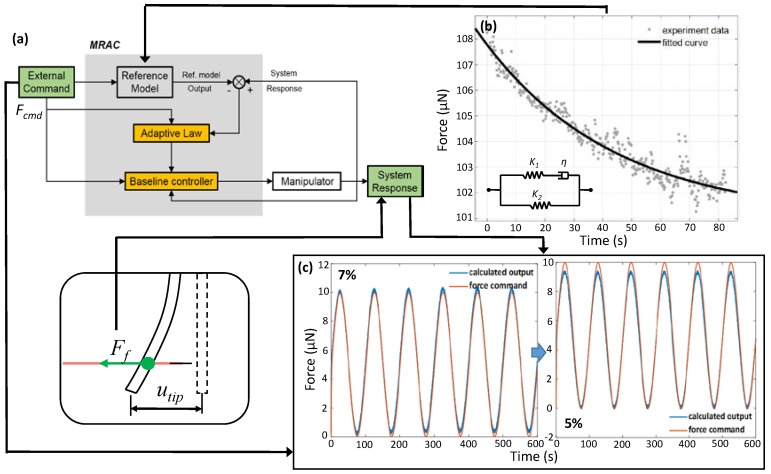
The stretching force control strategy. (**a**) Block diagram of the proposed MRAC method. (**b**) Experimental data fitted with SLS model based on 7% *w*/*v* GelMA microfiber. (**c**) Simulation results of MRAC with microfiber with 7% and 5.5% *w*/*v* GelMA microfiber.

**Figure 4 micromachines-13-01703-f004:**
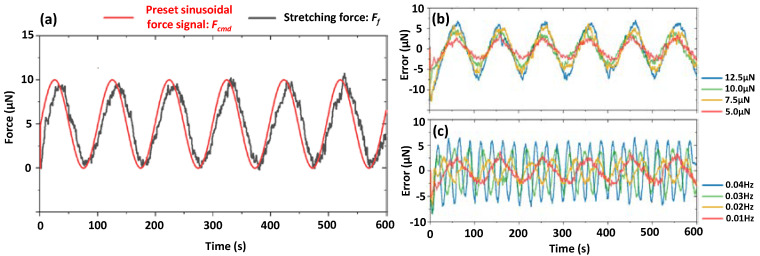
Tracking performance evaluation. (**a**) Time series of *F_f_* vs. *F_cmd_* (**b**) Time series of sine wave tracking error with different stress amplitudes. (**c**) Time series of sine wave tracking error with different frequencies.

**Figure 5 micromachines-13-01703-f005:**
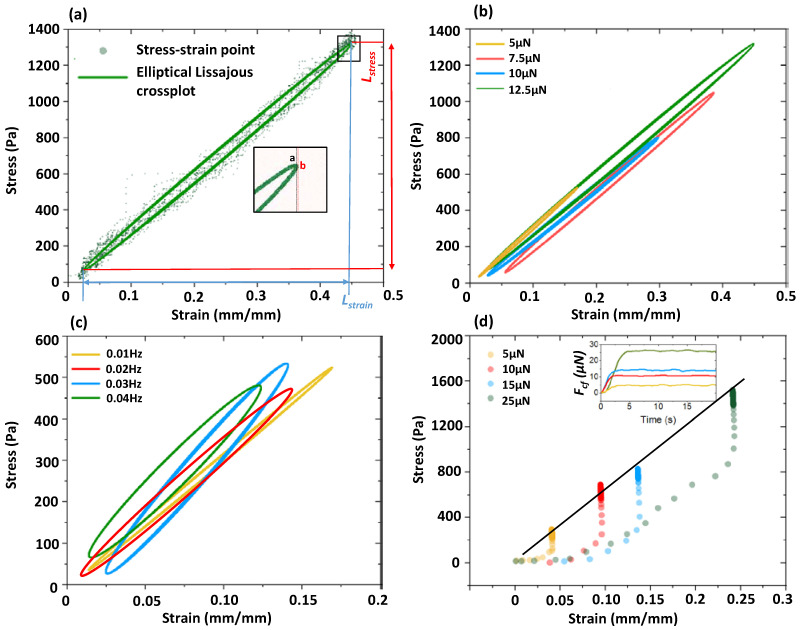
Analysis of stress-strain relationship. (**a**) Scatter plot from the original signals and elliptical Lissajous crossplot from the fitted signals to describe stress-strain loop. (**b**) Elliptical Lissajous crossplots under different stress amplitudes. (**c**) Elliptical Lissajous crossplots under different frequencies. (**d**) Stress-strain curve under different constant force loading. Insert is a time-dependent constant force control.

**Figure 6 micromachines-13-01703-f006:**
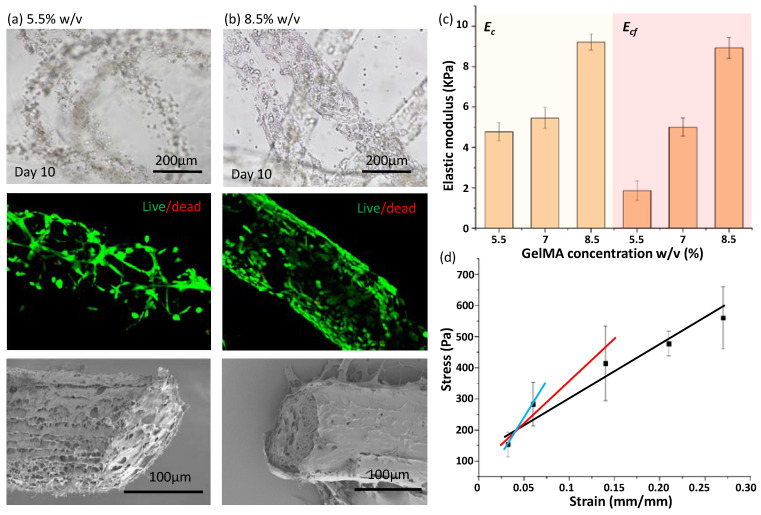
Reasonableness verification of E measurement. (**a**,**b**) Cell growth in 5.5% and 8.5% *w*/*v* GelMA microfibers, respectively. (Images of bright field, fluorescence for indicating live/dead cell, SEM for showing porous structure from top to bottom). (**c**) *E_c_* for cyclic force loading vs. *E_cf_* for constant force loading. (**d**) Stress-strain curves fitted from previous two (blue), three (red) sample data and whole sample data (black).

**Table 1 micromachines-13-01703-t001:** Fitted result by SLS model.

Parameter	$\lambda_{1}$	$\lambda_{2}$	τ	u	Goodness of Fit: R^2^
value	11.23	0.7629	44.94	9	0.948

**Table 2 micromachines-13-01703-t002:** Fitted results of GelMA microfiber with different concentrations.

Parameter	$\lambda_{1}$	$\lambda_{2}$	τ
7% *w*/*v* GelMA	11.23	0.7629	44.94
5.5% *w*/*v* GelMA	30.72	3.584	56.20

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
