# Peer review of "A Stretching Force Control-Based Cyclic Loading Method for the Evaluation of Mechanical Properties of Gelation Methacrylate (GelMA) Microfibers"

_micromachines, 2022, doi:10.3390/mi13101703_

Round 1

Reviewer 1 Report

In thisthe  paper, author propose a force control-based cyclic loading method for increasing the elastic modulus of GelMA microfibers. The work is interesting. They further showed cellular encapsulation using fluorescence staining and subsequent microscopy. Authos are suggested to address these concerns. 

1. I suggest studying the GelMA fiber viscoelastic properties if possible as supporting data. 

2. it seems that cells were mixed with the GelMA solution and prepared a fiber after that. If this is the case, did cells were not harmed by UV light during crosslinking? The authors should add more information in the material method section.

Author Response

Please check the response in the attachment 

Reviewer 2 Report

This manuscript reports a setup for measuring the mechanical properties of ultra-soft hydrogel fibers under static and dynamic loading. The force resolution (on the order of µN) is pretty impressive and can be a useful platform for characterizing the properties of ultra-soft hydrogel fibers. However, the quality of the current version needs to be improved to meet the standard for publication in the SCI-indexed micromachines journal. I have listed the major and minor comments below.

1.     On Page 2 line 46, the author claims one motivation for this work is the mechanical property of bulk GelMA hydrogel is difficult to describe exactly the mechanical property of GelMA hydrogel microfiber. To support this claim, the author should show how much difference between the mechanical property of bulk hydrogel and that of hydrogel fiber. For soft hydrogels, I feel there is little difference between elasticity and viscoelasticity, but a huge difference for fracture since fracture is sensitive to the size of defects.

2.     The authors should redraw the schematic to illustrate the working principle clearly. Fig. 2(a) is confusing and very difficult for readers to understand the mechanism for measuring force and displacement.

3.     There are some grammar issues. For example, on page 3 line 111, “two end of the microfiber is hanged” should be “two ends of the microfiber are hanged”.

4.     Some terminology is not well defined. For example, on page 2 line 81, “elliptical Lissajous crossplots” should be clearly defined.

5.     The formats of some variables are not consistent. For example, on page 7 line 15, the format of η is seemingly different from other variables.

6.     Hydrogel fiber is easy to dehydrate since the fiber size is small. How do the authors ensure the measured results are not affected by environmental humidity?

7.     The impact of this work is seemingly not that high. It is unclear how important the accurate measurement of modulus of hydrogel fibers is. In figure 6, the authors provide some information on cell growth. The question is, does it matter if one just uses the data measured from bulk hydrogels?

Author Response

(The authors gave the same response as above.)
